# Agronomic Evaluation and Chemical Characterization of *Lavandula latifolia* Medik. under the Semiarid Conditions of the Spanish Southeast

**DOI:** 10.3390/plants12101986

**Published:** 2023-05-15

**Authors:** Gustavo J. Cáceres-Cevallos, María Quílez, Gonzalo Ortiz de Elguea-Culebras, Enrique Melero-Bravo, Raúl Sánchez-Vioque, María J. Jordán

**Affiliations:** 1Research Group on Rainfed Agriculture for Rural Development, Department of Rural Development, Oenology and Sustainable Agriculture, Murcia Institute of Agri-Food and Environmental Research (IMIDA), La Alberca de las Torres, 30150 Murcia, Spain; gustavoj.caceres@carm.es (G.J.C.-C.); maria.quilez@carm.es (M.Q.); 2Instituto Regional de Investigación y Desarrollo Agroalimentario y Forestal de Castilla La Mancha (IRIAF), CIAF de Albaladejito, Carretera Toledo-Cuenca km 174, 16194 Cuenca, Spain; gonzaloo@jccm.es (G.O.d.E.-C.); emelerob@jccm.es (E.M.-B.); rsanchezv@jccm.es (R.S.-V.)

**Keywords:** *Lavandula latifolia*, agronomical yield, essential oil, phenolic profile, antioxidant activity

## Abstract

*Lavandula latifolia* is one of the main rainfed crops of aromatic and medicinal plants produced in Spain. As a global concern, the agronomic productivity of this aromatic crop is also threatened by the consequences of imminent climate change. On this basis, the study of the agronomic production of two drought-tolerant ecotypes, after three years of cultivations practices, constitutes the main objective of the present study. For this trial, clones of the two pre-selected ecotypes, along with clones from two commercial plants (control), were grown in an experimental plot. The main results confirmed an increase in biomass and essential oil production with plant age. The essential oil chemotype defined by 1,8-cineol, linalool, and camphor was maintained over time, but a decrease in 1,8-cineol in the benefit of linalool was detected. In the phenolic profile, 14 components were identified, with salvianic acid and a rosmarinic acid derivate being the main compounds quantified. These phenolic extracts showed potent in vitro antioxidant capacity, and after the second year of cultivation practices, both phenolic compounds and antioxidant capacity remained stable. Thus, under semiarid conditions, *L. latifolia* drought-tolerant ecotypes reach a good level of production after the second year of crop establishment.

## 1. Introduction

Global climate change associated with increasing temperatures will lead to more prolonged and severe drought episodes producing damages such as decreased growth and yield in commercial crops [1]. Therefore, selecting and studying drought-resistant aromatic-medicinal plants that can maintain growth and optimal agronomical yield in essential oils (EOs) under these conditions is a work of great importance [2]. In this sense, previous breeding studies performed by the authors allowed for the obtaining of drought-resistant elite individuals in early stages [3] that could be considered to analyze their EO yield and quality in Mediterranean semiarid field conditions. EOs are an important product of aromatic-medicinal plants (AMP), being found in exclusive families such as *Alliaceae*, *Apiaceae*, *Asteraceae*, *Myrtaceae*, *Poaceae*, *Rutaceae*, and *Lamiaceae* and being of outstanding value in the medicinal and pharmaceutical industries [4]. It is estimated that the global market for EOs is approximately USD 7.6 billion and could reach USD 15.1 billion by the year 2026 [5].

Within the *Lamiaceae* family, *Lavandula latifolia*, also known as spike lavender, is an aromatic shrub that grows wild in large areas of the Mediterranean region where the climate is warmer [6]. In fact, the edaphoclimatic conditions of the Spanish southeast have allowed for the establishment of extensive cultivated areas of this species, especially in regions such as Cuenca, Guadalajara, Albacete, and Murcia [7]. Thus, Spain has been considered as the largest producer of spike lavender EOs with approximately 150–200 t per year, being the most important commercialized EOs [8].

Furthermore, *Lavandula latifolia* is a commercially interesting AMP because its EOs have a high richness of monoterpenes, mainly linalool, 1,8-cineole (also called as eucalyptol), and camphor [9], for instance. EOs with lower levels in 1,8-cineole and camphor but higher levels of linalool are highly valued in the cosmetic and pharmaceutical industries [8]. However, as usual in AMP, the environmental conditions and plant genotype can affect the content of secondary metabolites in EOs [5].

For example, Fernández-Sestelo and Carrillo [8] mentioned that the EO yield of spike lavender is related to climate conditions, showing higher yields in rainy areas; moreover, Herraiz-Peñalver et al. [10] found that the chemical composition of EOs varied depending on the geographical situation, showing a similar composition in Spanish wild populations with identical edaphoclimatic conditions. In line with this, the nature of the soil may induce different molecule production secreted by microorganisms that stimulate and regulate the synthesis of EOs in medicinal plants [11]. In addition, EOs in lavender plants may vary between seasonal stages and plant parts [12].

Some studies on other plants belonging to the *Lamiaceae* family [13,14,15,16,17] showed that EO composition and yield are determined by environmental conditions, genotype, and growth stages, being necessary to analyze the plants according to their climatic conditions to ensure high yield and stable crops during the course of cultivation. 

Jordán et al. [18] described that the extraction process of EOs leaves a residue that is considered an important natural source of antioxidants; in fact, some authors mentioned that these extracts have diverse biological activities that depend on the main phenolic content in them [19,20,21]. Accordingly, the analysis of phenolic extracts of *Lavandula latifolia* may provide an innovative source of natural phenolic compounds.

Dobros et al. [22] suggested that phenolic extracts of flowers of some lavender species contain several phenols, such as flavonoids, hydroxybenzoic acids and hydroxycinnamic acids, including rosmarinic acid, caffeic acid, *p*-coumaric acid, ferulic acid, chlorogenic acid, sinapic acid, cinnamic acid, salvianolic acid, apigenin, and luteolin glycosides, among others. These phenolics are considered potent metabolites with various applications due to their biological activity, as mentioned above. Specifically, the richness in phenolic compounds in extracts has shown a considerable correlation with their antioxidant capacities [23,24]. Although spike lavender has been shown considerable economic interest for several applications in countless important areas, breeding information as well as the study of agronomical behavior growing in field conditions over time is still poorly known.

The importance of this study lies in the fact that in aromatic and medicinal plants, both ecological and agronomical conditions can improve the biomass and their essential oil yield per hectare [25]. At the same time, for future crop establishment, knowledge about the agronomic behavior and essential oil quality of different aromatic and medicinal plant accessions over time could be useful for the selection of biotypes [16]. In line with the above, and on the basis of the fact that knowledge regarding the growth and stability of *Lavandula latifolia* drought-resistant plants over time could be a suitable tool to improve the yield and production of this important crop, the main objective of the present study was to evaluate biomass production, essential oil yield and quality, and richness of phenolic compounds from extracts of pre-selected spike lavender ecotypes over three years of cultivation practices.

## 2. Results

### 2.1. Phytomass Production and Essential Oil Yield 

For the development of this study, the agronomic yield, expressed as phytomass and essential oil production, of four different ecotypes was determined. This research represents the continuation of a previous work, from which two individual ecotypes were preselected on the basis of their higher tolerance to drought stress [3]. Thus, the main objective of the present study was to continue with the characterization of these ecotypes (Es1 and Es2) by studying their agronomic yield when compared to two commercial spike lavender plants (Ec1 and Ec2).

Before starting the experimental assay, the four ecotypes were characterized to ensure that they had similar essential oil yield and chemical composition. Thus, differences in final yield could be related to agronomical behavior. The results concerning these initial values are shown in Table 1.

The results for phytomass production and essential oil yield, after three years of cultivation, showed a significant increase, even doubling production between the first and third year of agronomic assay (Figure 1) in all the ecotypes under study. The ecotypes used as control (Ec1 and Ec2) showed close results regarding fresh biomass production but not for essential oil yield. However, in the preselected ecotypes (Es1 and Es2), disparate results concerning these two parameters were found. The previous severe drought stress might have positively affected to Es2, but not Es1, which showed the lowest yield among the ecotypes studied. In fact, Es1 and Ec2 initially yielded the same essential oil content (2%, Table 1), but after cultivation practices, the agronomic behavior was different.

In this case, Es2 was the ecotype with the highest phytomass (fresh weight, FW) and essential oil production (17,738 kg FW/ha and 223 l/ha, respectively), followed by ecotype Ec1, even though their biomass production was the lowest in the second crop year (Figure 1a). This could lead to think that the essential oil synthesized by this species could be genetically predetermined, since an increase in biomass production is not always related to a significant improvement in essential oil yield, as can be observed for Ec1, Ec2, and Es1. This is a preliminary study, and further investigation is needed to confirm this statement. 

The ecotype that had the lowest yield through the years of cultivation was Es1, reaching in the last year an agronomic yield of 8034 kg FW/ha and an EO yield of 82 l/ha (Figure 1a,b).

### 2.2. Essential Oil Composition

A total of 32 major components were identified, which represented 95–100% of the volatile fraction of spike lavender essential oil (Table 2). At the quantitative level, plants maintained the chemotype after cloning and re-establishing in culture, with this relative composition being stable over time (Table 2). The chemotype was defined, as mentioned above, by three components, the monoterpenoid 1,8-cineole, and the terpenoids linalool and camphor, on the basis of the relative concentration at which these three major compounds were present in this volatile fraction.

Other compounds that showed averages above 1% in all ecotypes under study and that deserve to be commented on for their contribution to the pleasant spike lavender essential oil aroma are α-pinene, β-pinene, and α-terpineol. It is also interesting to note that in Ec2 and Es2, both with 1,8-cineol as the main component quantified, the terpenic alcohols γ-terpineol and terpinen-4-ol were detected at concentrations higher than 1%. As for sesquiterpenic hydrocarbons (Z)-α, bisabolene was quantified in Ec1, Es1, and Es2, while for Ec2, β-caryyophyllene and caryophyllene oxide were the two sesquiterpenes quantified at relative concentrations above 1%. 

Regarding the variation in the relative concentration of these volatile components over three years of cultivation practices, Ec1 showed little changes, being the most stable among the ecotypes studied. Thus, the components that define the chemotype in this plant, linalool/1,8-cineole/camphor, did not show statistically significant differences between the relative concentrations at which they were quantified over time. On the contrary, in Ec2, a detriment in the concentration of 1,8-cineole (from 62 to 53%) in favor of an increase in linalool (from 8.87 to 11%) and camphor (from 7 to 9%) was quantified. The volatile profile of Es1 also showed a reduction in 1,8-cineole after the third year of cultivation practice; however, in this case, linalool did not vary in statistically significant differences in its relative concentration between the first and third year of study. Ecotype Es2 showed an evolution closer to Ec1, since although a reduction in 1,8-cineol was quantified together with an increase in linalool concentration in the second year, no statistically significant differences were determined between the first and third year of cultivation. 

Minor components including the terpenic hydrocarbons sabinene (Ec1), myrcene (Ec1, Ec2, Es2), camphene (Ec2, Es2), α-terpinene and γ-terpinene (Es1), limonene (Es2), and α-terpinolene (Ec2, Es1, Es2), along with (*E*)-sabinene hydrate (Ec1), borneol (Ec2, Es2), γ-terpineol, terpinen-4-ol and α-terpineol (Ec2), and the sesquiterpenes β-caryophyllene (Ec1, Es1), germacrene (Ec1, Ec2), and viridiflorol (Ec1), increased their relative concentration over time. 

### 2.3. Phenolic Profile

Regarding the phenolic profile, chromatographic analysis (HPLC-DAD) allowed for the identification and quantification of 14 phenolic components (Table 3), with salvianic acid, rosmarinic acid derivate, luteolin-7-*O*-glucoronide, and salvianolic acid A being the major phenolic components quantified at the third year of cultivation practices for all the ecotypes under study.

Even the fact that spike lavender is a rainfed crop, the lack of water supply had a positive effect on the synthesis of phenolic components by this aromatic plant. Thus, except for *o*-coumaric acid, luteolin, and apigenin, all the phenolic components quantified had a considerable increase in their concentrations at the end of the experimental research. 

In this sense, salvianic acid was the component that modified its concentration to the greatest extent, being more evident in ecotype Es2 (Table 3). As expected, the variation in concentration was associated with the component under study; thus, the rosmarinic acid derivate, *p*-coumaric acid glycoside, ferulic acid hexoside, and rosmarinic acid experienced an increase from the first to the second year of cultivation, and then maintained their concentration level in the third year. However, for luteolin-7-*O*-glucoside and luteolin-7-*O*-glucoronide, this increase was determined just after the third year of cultivation practices. 

It is also interesting to note that, as defined for the essential oil volatile profile, ecotype Ec1 was the one that appeared most stable in terms of phenolic profile, since, unlike what was observed in the other ecotypes, namely, the derivative of rosmarinic acid, ferulic acid hexoside, *o*-coumaric acid, salvianolic acid A, and luteolin, no statistically significant differences were quantified between the three years of cultivation practices.

### 2.4. Antioxidant Activity

When analyzing the antioxidant activity of the methanolic extracts in three consecutive years of cultivation, it was evident that all ecotypes presented potent antioxidant activity in both DPPH and FRAP in vitro determinations (Figure 2).

Regarding the evolution of crops over time, all the ecotypes statistically increased their antioxidant potential in the third year compared to the first one. In addition, in Figure 2, it can be seen that Ec2 an Es2 presented the highest antioxidant activity for the DPPH and FRAP assays (94.1 ± 9.52 and 94.1 ± 9.52 µmol of Trolox equivalents (TE)/g DW 288.8 ± 44.66 and 281.4 ± 21.69 µmol Fe^+2^ g/DW, respectively). However, between the second and third cropping year, all ecotypes did not show statistical differences in antioxidant activity, remaining stable from the second year onwards.

To evaluate the relationships between phenolic compounds, correlation coefficients were measured, showing a strong significant correlation (*p* < 0.05) in DPPH and FRAP with *p*-coumaric acid glycoside (r = 0.88 and r = 0.91, respectively). Similarly, salvianolic acid A correlated highly with the DPPH method (r = 0.828), whereas the FRAP method correlated strongly with luteolin-7-glucuronide (r = 0.895).

## 3. Discussion

Variations in the agronomic behavior of aromatic plants are associated not only with the species under study, but also, among others, with environmental conditions, plant age, and the cultivation practices [26]. Therefore, to produce high quality spike lavender plant material with stable and reproducible agronomic yield, it is crucial to evaluate the evolution of the agronomic production along with its chemical characterization over time. 

Thus, the evaluation of four ecotypes during three years of cultivation practices showed that, although phytomass production and essential oil yield increased in the third year of cultivation practices, the agronomic behavior was ecotype dependent. In the case of the two preselected ecotypes (Es1 and Es2), which were previously subjected to two severe drought episodes, a priming effect could occur in their DNA [27], which could explain the differences found between their respective agronomic yield. At the same time, when comparing phytomass production and essential oil yield over time, no direct proportionality between both parameters was observed. Variations in biomass production between the second and third year of cultivation were lower than the observed increase in essential oil yield. This could be justified by environmental conditions. As shown in Figure 3, the differences between the second and third year were related to the level of potential evapotranspiration, being lower in the third year, which could explain the higher level of essential oil yield obtained. Previous studies by Herráiz-Peñalver et al. [10] and Fernández-Sestelo and Carrillo [8] described variations in the essential oil content of spike lavender depending on annual climatic conditions. For these authors, in wild populations of spike lavender harvested from different bioclimatic zones of Spain, a positive correlation between rainfall and essential oil content was determined. 

According to the scientific literature consulted, and to the best of our knowledge, no studies related to the agronomic behavior of spike lavender have been previously carried out. Therefore, the discussion of the results is based on those obtained with other *Labiateae* species. Thus, contrary to the results determined for spike lavender, Tuttolomondo et al. [16] reported, for *Salvia sclarea*, a decrease (by half) in biomass and essential oil yield after two years of cultivation practices. For these authors, both factors, environmental and genetic conditions, could play an important role in this production detriment. However, Najar et al. [28] for *Thymus vulgaris* L. described an agronomic behavior close to that reported for spike lavender. According to these researchers, after three years of cultivation practices, thyme experimented an increase in biomass production, but not for essential oil yield, which was maintained over time. 

As for the essential oil volatile profile, the chemotype described in the four ecotypes under study, defined by linalool, 1,8-cineol, and camphor, was previously described in individual plants harvested from natural populations of *L. latifolia* from the Eastern Iberian Peninsula [8,10,15,29]. 

Variations in essential oil composition over time showed a different pattern of behavior depending on the chemotype studied. Thus, the evolution of the volatile profile in the ecotypes used as control (Ec1 and Ec2) could point to the fact that genetic factors, rather than environmental conditions, could be directly related to the response of the secondary metabolism of this species, as reported by Farías et al. for four clones of *Origanum* spp. from the Argentine Littoral region [25]. This could explain why Ec1 and Ec2 exhibited different evolutions after three years of cultivation practices under the same edaphoclimatic conditions. 

According to the climatic conditions shown in Figure 3 in 2020, there was a higher annual precipitation compared to the other two years under study. However, in the third year, the evapotranspiration of the area was lower. Therefore, the plants in 2021 were subjected to a greater lack of water availability. Under this situation, these ecotypes, as a priming effect, could respond to water scarcity by increasing linalool and camphor and reducing 1,8-cineol. 

Contrary to this, Muñoz-Bertomeu et al. [29] indicated that after a prospection of individual spike lavender plants located in different bioclimatic belts of eastern Spain, milder temperature conditions and higher levels of precipitation (Supra-Mediterranean belt) were found to favor linalool synthesis, decreasing the relative concentration of 1,8-cineol and camphor in the volatile fraction of these plants. Subsequently, Fernandez-Sestelo and Carrillo [8] in their study on the environmental effects on essential oil yield and composition of spike lavender wild populations also found a significant negative correlation between the relative concentrations of 1,8-cineol, linalool, and camphor. The differences between our results and those reported by these researchers could be attributed to the large chemical variability described for this species. According to Muñoz-Bertomeu et al. [29], this intraspecific variability has not only been described among wild populations, but also among individual plants. As indicated above, for the development of this study, clones of four different ecotypes were cultivated for study. Thus, variations among clones of the same ecotype should yield more conclusive results regarding the evolution of secondary metabolism in this species. Our results showed that in three of the four ecotypes, there was an increase in the relative concentration of linalool and camphor to the detriment of 1,8-cineol after three years of cultivation practices under the same edaphoclimatic conditions. 

Continuing with the secondary metabolism of these plants, and related to the phenolic profile, at present, there is scarce information on these bioactive components extracted from the distillation residues of this aromatic species. Researchers, such as Mendez-Tovar et al. [30], described rosmarinic acid, luteolin, and apigenin as the main phenols identified and quantified in *L. latifolia* wild populations. Recently, Dobros et al. [31] published a detailed review on the phytochemical profile of plants belonging to the *Lavandula* genus. For these authors, rosmarinic acid was the most abundant compound quantified in all the species covered in their review. Although at the qualitative level, all the phenolic compounds identified in the four ecotypes under study have been previously reported in the *Lavandula* genus, differences were observed in the concentration at which they were quantified. Variations in the qualitative and quantitative phenolic profile can be attributed to different factors, including genetic (inter and intraspecific variability) and physiological and development stages of the plants [32]. Thus, depending on the geographical origin of the plant material, differences in agronomic behavior and chemical composition are expected. In fact, it is widely known that phenolic compounds have a great capacity to perform multiple functions in plants exposed to a wide range of abiotic stresses [33].

In the Mediterranean area, plants are exposed to multiple severe climate conditions, which could explain the evolution of the phenolic profile of spike lavender over the three years of cultivation practices. A possible elucidation of this statement could be related to the fact that these components, including phenolic glucosides, hydroxycinnamic acid derivates, and flavoinoids, are involved in secondary cell wall thickening [34] that confers drought tolerance to plants. This cell reinforcement coupled with their chemical functions confer to these phenolic components an important role in the resistance of the plants to oxidative stress [35]. Thus, under the semiarid conditions of the Spanish Southeast, spike lavender experienced a significant increase in the concentration of rosmarinic acid derivate, *p*-coumaric acid glycoside, ferulic acid hexoside, rosmarinic acid, luteolin-7-*O*-glucoside, and luteolin-7-*O*-glucoronide. The components showed a positive correlation with both in vitro antioxidant test assayed and could help the plant to overcome the adverse climatic conditions of southeastern Spain. 

## 4. Materials and Methods

### 4.1. Crop Experimental Design and Plant Material

The present study was carried out in an experimental area of IMIDA at “Chaparral” (38°06′39.6″ N 1°40′50.0″ W, and 400 m above sea level) in Murcia (Spain). Soil conditions in the first 30 cm of the cultivation area can be defined as clay loam (silt (38%), sand (30%), and clay (33%)), with an alkaline pH (8.07), a percentage of soil saturation of 36%, and electrical conductivity of 0.85 dS/m. This experimental station has semiarid climatic conditions, as shown in Figure 3, in relation to the monthly average rainfall, temperature, and evapotranspiration throughout this experiment. 

**Figure 3 plants-12-01986-f003:**
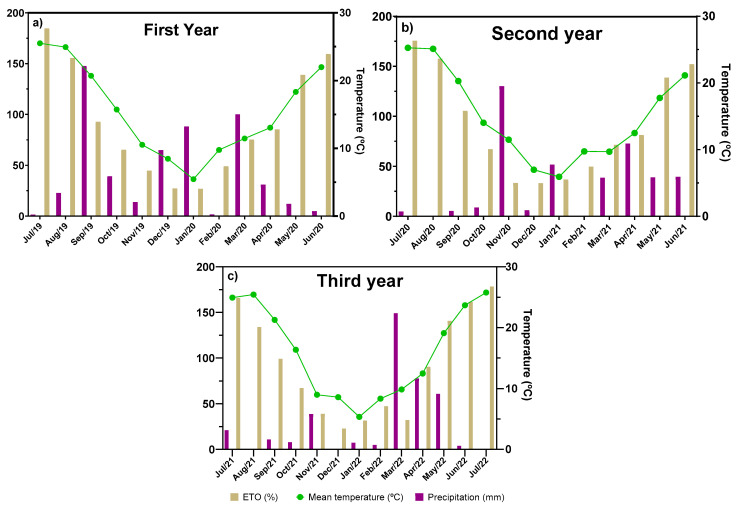
Climatic data of the experimental area of the first, second, and third year crop ((**a**,**b**,**c**), respectively).

An assay with three randomized blocks and ten experimental replications per ecotype and block was designed. Each replication area has 10 m^2^, with a planting distance of 0.5 × 1 m, making a total of 120 plants, with a density of 200 plants/100 m^2^. 

The plant material used as control (Ec1 and Ec2) was obtained from a local AMP producer. In this case, two individual plants were chosen on the basis of their essential oil yield and chemotype composition. The other two ecotypes (Es1 and Es2) came from the IMIDA germplasm bank, previously selected as drought-resistant plants [3]. 

The plants under study were multiplied by in vitro culture following the protocol described by Cáceres-Cevallos et al. [36]. A total of 30 clones of each ecotype were produced. Subsequently, all plantlets were acclimatized and placed in greenhouse conditions for three months until transplanting to the experimental area in April 2019. 

Spike lavender is an undemanding crop in nutrients, and therefore just a base dressing was applied at pre-planting. Weed control from the first year of planting was controlled mechanically between rows while manually on the rows. 

To evaluate phytomass production and essential oil yield and quality over three years of cultivation, three consecutive harvests were made during the summers of 2020, 2021, and 2022, when the plant material showed a phenological stage of full blooms and the beginning of fructification (which occurred in July, in the third week in 2020 and in the second week in 2021 and 2022). The fresh plant material was weighed immediately after harvesting, and a sample of 3 kg was kept for the determination of dry matter production. For this purpose, the plant material was dried in a forced-air dryer at 35 °C for 48 h, until it reached a constant weight. The dry matter obtained per 100 m^2^ of cultivation area was determined by extrapolation of the dry matter weight of 3 kg of fresh plant material to the total production in 100 m^2^ of cultivation area [37]. 

Before essential oil extraction, floral stems from four clones per ecotype and block were dried in a forced-air dryer at 35 °C for 48 h, until they reached a constant weight.

### 4.2. Essential Oil Extraction

Essential oils were extracted according to the protocol described by the European Pharmacopoeia [38]; briefly, the dry floral stems of four clones per ecotype and block were subjected to hydrodistillation for 3 h using a Clevenger-type apparatus. The oil obtained was dried with anhydrous sodium sulphate and stored in amber vials at −20 °C until chromatographic analysis. The percentage yield of EOs was calculated as volume (milliliters) of essential oil per 100 g of distilled floral stems. Essential oil yield expressed as l/ha was calculated considering the weight per plant, the percentage of essential oil, and the density of plantation.

### 4.3. Gas Chromatography–Mass Spectrometry (GC-MS) Analysis

For the qualitative and quantitative analysis of the volatile profile of the essential oil, a method described by Jordán et al. [17] was used. Briefly, the essential oil samples (0.1 µL) were subjected to analysis by GC-MS. A 6890 N gas chromatograph (GC) (Palo Alto, CA, USA) equipped with a 30 m × 0.25 mm i.d. HP-5 (5% cross-linked phenyl-methyl siloxane) column with 0.25 µm film thickness was used. The stationary phase was supplied by Agilent Technologies (Palo Alto, CA, USA). Helium was used as the carrier gas (constant pressure, β-ionone eluting at 27.6 min), and the split ratio was set at 100:1. The GC was linked to an Agilent model 5972 inert mass spectrometry detector. The initial oven temperature was set at 60 °C, then increased at 2.5 °C/min to 155 °C, and finally raised to 250 °C at a rate of 10 °C * min^−1^; the injection port and transfer line to the mass selective detector were kept at 250 °C and 280 °C, respectively. Individual peaks were identified by retention times and retention indices (relative to C6-C17 n-alkanes), compared with those of known compounds, and by comparison of mass spectra using the NBS75K library (U.S. National Bureau of Standards, 2002) and spectra obtained from the standard. The percentage compositions of samples were expressed as a function of the area of the chromatographic peaks using the total ion current.

### 4.4. Extraction of Phenolic Compounds

After the distillation procedure, the distilled plant material was dried in a forced-air dryer at 35 °C for 48 h to reach a constant weight, and then ground to pass through a 2 mm mesh. Phenolic compounds were extracted following the method described by Jordán et al. [18], with some modifications. Briefly, 500 mg of dried samples were extracted using 150 mL of methanol in a Soxhlet extractor (B-811) (Buchi, Flawil, Switzerland) for 2 h under nitrogen atmosphere. The methanolic extracts (ME) were brought to dryness at 40 °C under vacuum conditions in an evaporator system (SyncorePolyvap R-96) (Buchi, Flawil, Switzerland). The residue was re-dissolved in methanol and made up to 5 mL. The final extracts were stored in vials at −80 °C until their corresponding HPLC-DAD analyses.

### 4.5. HPLC-DAD Analysis

For qualitative and quantitative analysis, a method adapted from Jordán et al. [17] was followed, using a reverse-phase Zorbax SB-C18 column (4.6 mm × 250 mm, 5 µm pore size, Agilent Technologies, Santa Clara, CA, USA) with a guard column (Zorbax SB-C18 4.6 mm × 125 mm, 5 µm pore size, Agilent Technologies, Santa Clara, CA, USA) at ambient temperature. The extracts were passed through a 0.45 µm filter (Millipore SAS, Molsheim, France), and 20 µL was injected into a HPLC 1200 (Agilent Technologies, Santa Clara, CA, USA) system equipped with a G1311A quaternary pump and G1315A photodiode array UV-Vis detector.

The mobile phase was acetonitrile (B) and acidified water containing 0.05% formic acid (A). The gradient used was 0 min, 5% B; 10 min, 15% B; 30 min, 25% B; 35 min, 30% B; 50 min, 55% B; 55 min, 90% B; 57 min, 100% B, held for 10 min before returning to the initial conditions. The flow rate was 1.0 mL × min^−1^, and the wavelengths of detection were programmed at 280 and 330 nm. Identification of the phenolic components was made by comparing retention times and spectra with those of commercially available standard compounds. To quantify, linear regression models were determined using standard dilution techniques. Phenolic compound contents were expressed as mg per g of dry weight.

### 4.6. Antioxidant Capacity (FRAP and DPPH Assays)

FRAP was measured as the ability of the methanolic extract to reduce ferric ions, following the method described by Benzie and Strain [39]. Briefly, FRAP reagent was made with 300 mM acetate buffer (pH 3.6), 10 mM TPTZ (2,4,6-tripyridyl-*s*-triazine) prepared with 40 mM HCl, and 20 mM FeCl_3_. The three solutions were mixed in a proportion of 10:1:1 (*v*/*v*/*v*). An aliquot of 40 μL was combined with 1.2 mL of FRAP reagent, after which the aliquot was incubated for 30 min in the dark at 37 °C. All samples were measured spectrophotometrically at 593 nm. Fresh working solutions of known Fe (II) concentrations (FeSO_4_·7H_2_O) of 0.1–1 mM were used for the calibration curve. Results were expressed as µmol of Fe^+2^/g of dry plant. The ability of methanolic extracts to scavenge DPPH free radicals was determined following the method described by Brand-Williams et al. [40]. Briefly, 500 μL of aliquots was added to 1 mL of methanolic solution DPPH^•^ (0.1 mM). The samples were incubated for 20 min at room temperature in the dark and then measured in spectrophotometer (Shimadzu UV-2401PC, Kyoto, Japan) at 517 nm. The absorbance was measured against a blank of 500 μL of methanol mixed with 1 mL of DPPH^•^ solution. The calibration curve of Trolox used was in the range of 100–450 µM. Results were expressed as µmoles of Trolox equivalents/g of dry plant.

### 4.7. Statistical Analyses

Data are shown as mean ± SE. Data were analyzed by one-way analysis of variance (ANOVA), verifying previous homoscedasticity by Levene’s test. Fisher LSD test was used to compare means at *p* < 0.05. Correlations in treatments were measured by Pearson’s correlation coefficients. All statistical analyses were estimated by the software STATGRAPHICS Centurion XVI.I.

## 5. Conclusions

The agronomical behavior and the evolution of the essential oil and the phenolic profiles of clones of four ecotypes, cultivated for three years in the semiarid conditions of the Spanish Southeast, were ecotype dependent, reaching a stable level of production in the second year of cultivation. Under the same edaphoclimatic conditions, the evolution over time in terms of phytomass and essential oil yield suggested that the oil synthesized by this species could be genetically predetermined, since an increase in biomass production is not always related to a significant improvement in essential oil yield. Regarding the volatile profile of the essential oil, the secondary metabolism of this species, in three of the four ecotypes, seemed to redirect the synthesis of 1,8-cineol to linalool and camphor in response to the harsh Mediterranean climatic conditions. For the phenolic profile, and consequently for the antioxidant capacity of their extracts, an increase was observed between the first and second year but remained stable in the third. This tendency could suggest that after episodes of water shortage, the species could maintain this antioxidant system active over time. However, more studies, over a longer period, are needed to confirm these statements. Knowledge of this agronomic behavior is of utmost importance for the development of future *Lavandula latifolia* breeding programs.

## Figures and Tables

**Figure 1 plants-12-01986-f001:**
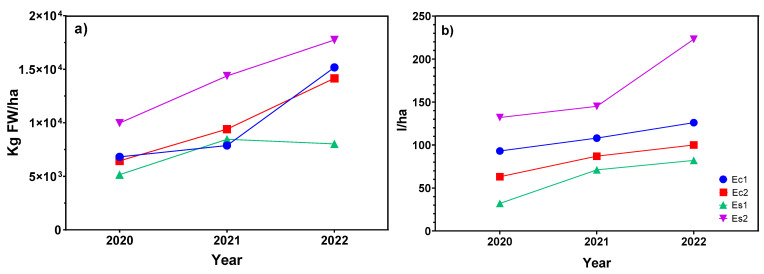
Evaluation of (**a**) agronomic yield and (**b**) essential oil yield in *Lavandula latifolia* during three consecutive years.

**Figure 2 plants-12-01986-f002:**
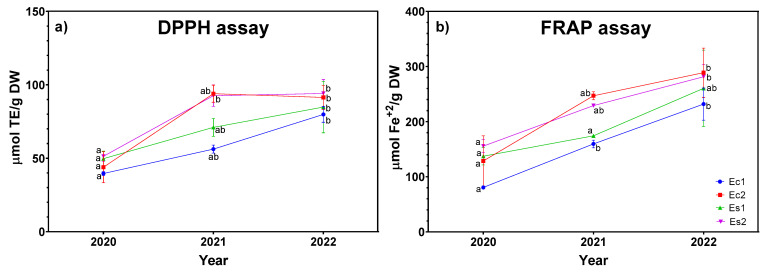
Antioxidant activity of *Lavandula latifolia* in three consecutive crop years by methods (**a**) DPPH assay and (**b**) FRAP assay. Different letters indicate a significant difference between years at *p* < 0.05.

**Table 1 plants-12-01986-t001:** Essential oil yield and chemotype definition of the four ecotypes under study.

	Ec1	Ec2	Es1	Es2
Essential oil yield (%)	3.0	2.2	2.3	2.9
Chemotype	linalool/1,8-cineol/camphor			
Relative concentration (%)	48/32/9	10/46/9	40/35/9	26/45/9

**Table 2 plants-12-01986-t002:** Essential oil volatile profile of *Lavandula latifolia* over three years of cultivation practices.

Components Relative Concentration (%)	RI	Year	Ec1	Ec2	Es1	Es2
α-Pinene	963	2020	0.9 ± 0.20	1.9 ± 0.22	2.1 ± 0.18	1.5 ± 0.20
		2021	0.9 ± 0.06	1.7 ± 0.20	1.5 ± 0.37	1.5 ± 0.12
		2022	1.0 ± 0.09	2.0 ± 0.20	1.4 ± 0.27	1.6 ± 0.10
Camphene	971	2020	0.2 ± 0.11	0.2 ± 0.03 ^a^	0.2 ± 0.04 ^a^	0.2 ± 0.02 ^a^
		2021	0.3 ± 0.01	0.3 ± 0.02 ^b^	0.3 ± 0.06 ^ab^	0.3 ± 0.05 ^b^
		2022	0.3 ± 0.11	0.4 ± 0.01 ^b^	0.3 ± 0.02 ^b^	0.4 ± 0.01 ^b^
Sabinene	987	2020	0.4 ± 0.02 ^a^	1.1 ± 0.12	0.7 ± 0.03	0.8 ± 0.05
		2021	0.5 ± 0.02 ^a^	1.0 ± 0.04	0.6 ± 0.12	0.8 ± 0.04
		2022	0.6 ± 0.02 ^b^	1.2 ± 0.05	0.7 ± 0.07	0.9 ± 0.09
β-Pinene	989	2020	1.4 ± 0.18	3.5 ± 0.34 ^b^	2.8 ± 0.19 ^b^	2.7 ± 0.26
		2021	1.3 ± 0.07	2.8 ± 0.17 ^a^	1.9 ± 0.33 ^a^	2.3 ± 0.11
		2022	1.6 ± 0.18	3.5 ± 0.26 ^b^	2.1 ± 0.26 ^a^	2.7 ± 0.14
1-Octen-3-ol	991	2020	0.0 ± 0.01	0.1 ± 0.02	0.0 ± 0.01	0.0 ± 0.00
		2021	0.0 ± 0.01	0.1 ± 0.00	0.0 ± 0.01	0.0 ± 0.01
		2022	0.1 ± 0.01	0.1 ± 0.05	0.1 ± 0.05	0.0 ± 0.01
Myrcene	998	2020	0.3 ± 0.03 ^a^	0.7 ± 0.09 ^a^	0.5 ± 0.01	0.5 ± 0.06 ^a^
		2021	0.4 ± 0.02 ^a^	0.7 ± 0.06 ^a^	0.4 ± 0.07	0.6 ± 0.0 ^ab^
		2022	0.5 ± 0.03 ^b^	0.9 ± 0.03 ^b^	0.5 ± 0.04	0.6 ± 0.00 ^b^
α-Terpinene	1017	2020	0.1 ± 0.06	0.2 ± 0.06	0.1 ± 0.00 ^a^	0.2 ± 0.06
		2021	0.1 ± 0.01	0.2 ± 0.05	0.1 ± 0.02 ^a^	0.2 ± 0.03
		2022	0.2 ± 0.06	0.2 ± 0.05	0.2 ± 0.02 ^b^	0.3 ± 0.01
*p*-Cymene	1024	2020	0.1 ± 0.03	0.1 ± 0.02	0.1 ± 0.00	0.1 ± 0.04
		2021	0.1 ± 0.01	0.1 ± 0.00	0.1 ± 0.02	0.1 ± 0.01
		2022	0.1 ± 0.03	0.1 ± 0.01	0.1 ± 0.03	0.1 ± 0.01
Limonene	1027	2020	0.5 ± 0.27	1.3 ± 0.08 ^b^	0.8 ± 0.16	0.7 ± 0.15
		2021	0.5 ± 0.06	1.0 ± 0.05 ^a^	0.6 ± 0.14	0.7 ± 0.06
		2022	0.8 ± 0.27	1.5 ± 0.05 ^c^	0.8 ± 0.04	0.7 ± 0.08
1,8-Cineole	1030	2020	31.8 ± 2.52	62.1 ± 0.90 ^c^	43.2 ± 0.10 ^c^	49.5 ± 2.85 ^b^
		2021	27.1 ± 1.00	45.5 ± 0.97 ^a^	31.7 ± 0.82 ^a^	39.9 ± 1.31 ^a^
		2022	30.0 ± 3.01	53.3 ± 1.08 ^b^	36.3 ± 2.01 ^b^	45.7 ± 3.86 ^b^
(*E*)-β-Ocimene	1035	2020	0.2 ± 0.08	0.2 ± 0.02 ^a^	0.2 ± 0.03	0.2 ± 0.03
		2021	0.3 ± 0.01	0.3 ± 0.03 ^b^	0.2 ± 0.01	0.2 ± 0.02
		2022	0.2 ± 0.07	0.2 ± 0.02 ^a^	0.2 ± 0.04	0.1 ± 0.01
γ-Terpinene	1053	2020	0.2 ± 0.04	0.5 ± 0.02 ^a^	0.2 ± 0.00 ^a^	0.3 ± 0.09
		2021	0.2 ± 0.02	0.4 ± 0.03 ^b^	0.2 ± 0.03 ^a^	0.3 ± 0.05
		2022	0.2 ± 0.04	0.5 ± 0.02 ^a^	0.3 ± 0.02 ^b^	0.4 ± 0.02
(Z)-Sabinenehydrate	1061	2020	0.1 ± 0.06 ^a^	0.3 ± 0.04	0.3 ± 0.13	0.3 ± 0.10
		2021	0.3 ± 0.06 ^b^	0.5 ± 0.12	0.4 ± 0.01	0.4 ± 0.12
		2022	0.3 ± 0.06 ^b^	0.5 ± 0.10	0.2 ± 0.06	0.2 ± 0.12
α-Terpinolene	1080	2020	0.1 ± 0.08	0.1 ± 0.01 ^a^	0.1 ± 0.00 ^a^	0.1 ± 0.02 ^a^
		2021	0.2 ± 0.01	0.2 ± 0.02 ^b^	0.2 ± 0.03 ^b^	0.2 ± 0.03 ^b^
		2022	0.2 ± 0.08	0.3 ± 0.02 ^b^	0.2 ± 0.01 ^b^	0.2 ± 0.14 ^b^
Linalool	1094	2020	47.5 ± 0.98	8.7 ± 0.82 ^a^	34.8 ± 0.32 ^a^	26.5 ± 4.32
		2021	47.3 ± 0.66	18.7 ± 0.76 ^c^	41.3 ± 3.14 ^b^	30.5 ± 0.49
		2022	43.5 ± 0.98	11.0 ± 1.08 ^b^	36.1 ± 0.96 ^a^	25.8 ± 3.49
α-Campholenal	1118	2020	0.1 ± 0.01	0.1 ± 0.02	0.1 ± 0.01 ^a^	0.1 ± 0.01
		2021	0.1 ± 0.02	0.1 ± 0.04	0.2 ± 0.00 ^b^	0.1 ± 0.06
		2022	0.1 ± 0.01	0.1 ± 0.01	0.1 ± 0.02 ^b^	0.1 ± 0.01
Pinocarveol	1130	2020	0.1 ± 0.07	0.2 ± 0.04	0.2 ± 0.03	0.2 ± 0.05
		2021	0.1 ± 0.03	0.2 ± 0.02	0.2 ± 0.01	0.1 ± 0.05
		2022	0.1 ± 0.01	0.1 ± 0.04	0.0 ± 0.01	0.0 ± 0.01
Camphor	1136	2020	9.2 ± 1.43	7.0 ± 0.57 ^a^	5.9 ± 1.48	7.7 ± 0.15 ^a^
		2021	9.5 ± 0.30	10.2 ± 0.42 ^c^	8.6 ± 0.98	10.4 ± 1.18 ^b^
		2022	8.7 ± 0.45	9.2 ± 0.09 ^b^	9.1 ± 1.30	9.5 ± 0.30 ^b^
(*E*)-Pinocarveol acetate	1138	2020	0.1 ± 0.01	0.2 ± 0.04 ^b^	0.2 ± 0.03	0.2 ± 0.02
		2021	0.1 ± 0.01	0.2 ± 0.02 ^b^	0.3 ± 0.03	0.1 ± 0.02
		2022	0.0 ± 0.01	0.1 ± 0.04 ^a^	0.3 ± 0.01	0.0 ± 0.01
Pinocarvone	1156	2020	0.1 ± 0.01	0.1 ± 0.04 ^b^	0.1 ± 0.02 ^a^	0.1 ± 0.03
		2021	0.1 ± 0.01	0.1 ± 0.02 ^b^	0.2 ± 0.01 ^b^	0.1 ± 0.01
		2022	0.0 ± 0.01	0.0 ± 0.01 ^a^	0.0 ± 0.01 ^a^	0.0 ± 0.01
Borneol	1159	2020	0.7 ± 0.15	0.4 ± 0.09 ^a^	0.4 ± 0.07 ^a^	0.5 ± 0.03 ^a^
		2021	0.8 ± 0.06	0.8 ± 0.03 ^b^	0.8 ± 0.11 ^b^	0.8 ± 0.06 ^b^
		2022	0.9 ± 0.19	0.8 ± 0.01 ^b^	0.8 ± 0.14 ^b^	0.8 ± 0.07 ^b^
γ-Terpineol	1161	2020	0.6 ± 0.03	1.2 ± 0.05 ^b^	0.7 ± 0.02 ^a^	1.0 ± 0.10 ^a^
		2021	0.6 ± 0.02	1.1 ± 0.02 ^a^	0.7 ± 0.00 ^a^	0.9 ± 0.01 ^a^
		2022	0.7 ± 0.08	1.3 ± 0.04 ^c^	0.9 ± 0.04 ^b^	1.1 ± 0.03 ^b^
Terpinen-4-ol	1172	2020	0.6 ± 0.07	1.2 ± 0.03 ^b^	0.7 ± 0.00	0.9 ± 0.21
		2021	0.5 ± 0.05	1.0 ± 0.08 ^a^	0.6 ± 0.03	0.9 ± 0.10
		2022	0.6 ± 0.07	1.1 ± 0.04 ^b^	0.8 ± 0.08	1.1 ± 0.06
α-Terpineol	1187	2020	1.6 ± 0.16	3.6 ± 0.10 ^b^	2.2 ± 0.15	2.7 ± 0.11
		2021	1.6 ± 0.07	3.2 ± 0.05 ^a^	1.9 ± 0.14	2.6 ± 0.04
		2022	1.7 ± 0.16	3.5 ± 0.10 ^b^	2.1 ± 0.05	2.7 ± 0.14
Myrtenal	1192	2020	0.2 ± 0.02	0.3 ± 0.12	0.2 ± 0.03	0.2 ± 0.05 ^a^
		2021	0.2 ± 0.02	0.3 ± 0.03	0.3 ± 0.01	0.2 ± 0.02 ^a^
		2022	0.2 ± 0.02	0.3 ± 0.02	0.3 ± 0.03	0.3 ± 0.02 ^b^
Myrtenol	1193	2020	0.1 ± 0.01	0.0 ± 0.01	0.0 ± 0.01	0.0 ± 0.01
		2021	0.1 ± 0.01	0.1 ± 0.06	0.1 ± 0.04	0.1 ± 0.01
		2022	0.1 ± 0.01	0.2 ± 0.02	0.1 ± 0.02	0.1 ± 0.01
β-Caryophyllene	1423	2020	0.1 ± 0.01 ^a^	1.3 ± 0.18 ^b^	0.2 ± 0.00 ^a^	0.3 ± 0.08 ^a^
		2021	0.4 ± 0.06 ^c^	0.2 ± 0.01 ^a^	0.5 ± 0.07 ^b^	0.5 ± 0.08 ^b^
		2022	0.3 ± 0.02 ^b^	1.5 ± 0.13 ^b^	0.4 ± 0.01 ^b^	0.3 ± 0.05 ^a^
β-Farnesene	1468	2020	0.1 ± 0.06	0.0 ± 0.01 ^a^	n.d.	0.0 ± 0.00
		2021	0.1 ± 0.02	0.2 ± 0.06 ^b^	n.d.	0.1 ± 0.03
		2022	0.1 ± 0.02	0.0 ± 0.01 ^a^	n.d.	0.0 ± 0.00
Germacrene D	1485	2020	0.1 ± 0.04 ^a^	0.1 ± 0.04 ^a^	0.1 ± 0.06	0.2 ± 0.03 ^a^
		2021	0.4 ± 0.05 ^c^	0.4 ± 0.40 ^c^	0.5 ± 0.07	0.4 ± 0.07 ^b^
		2022	0.3 ± 0.01 ^b^	0.5 ± 0.07 ^b^	0.2 ± 0.05	0.2 ± 0.05 ^a^
(*Z*)-α-Bisabolene	1555	2020	1.3 ± 0.23	0.0 ± 0.01 ^a^	1.6 ± 0.76	1.0 ± 0.04
		2021	1.9 ± 0.35	0.2 ± 0.06 ^b^	2.0 ± 0.04	1.2 ± 0.28
		2022	2.1 ± 0.31	0.0 ± 0.01 ^a^	1.9 ± 0.44	0.7 ± 0.26
Caryophyllene oxide	1591	2020	0.4 ± 0.07	0.9 ± 0.12 ^ab^	0.1 ± 0.01 ^a^	0.3 ± 0.14
		2021	0.4 ± 0.03	1.2 ± 0.14 ^b^	0.4 ± 0.07 ^b^	0.5 ± 0.07
		2022	0.4 ± 0.04	0.7 ± 0.23 ^a^	0.1 ± 0.03 ^a^	0.4 ± 0.06
Viridiflorol		2020	0.2 ± 0.02 ^a^	0.9 ± 0.10	0.3 ± 0.18	0.4 ± 0.17
		2021	0.9 ± 0.10 ^b^	0.9 ± 0.13	0.7 ± 0.08	0.6 ± 0.13
		2022	0.9 ± 0.11 ^b^	0.8 ± 0.07	0.6 ± 0.13	0.5 ± 0.14

Different letters indicate a significant difference between years per component at *p* < 0.05.

**Table 3 plants-12-01986-t003:** Phenolic profile analysis of four spike lavender ecotypes.

Phenolic Compounds mg *g of Dry Weight (DW)^−1^	Year	Ec1	Ec2	Es1	Es2
	2020	0.3 ± 0.05 ^a^	0.6 ± 0.03 ^a^	0.7 ± 0.19 ^a^	0.6 ± 0.19 ^a^
Salvianic acid	2021	1.2 ± 0.48 ^b^	0.6 ± 0.22 ^a^	0.7 ± 0.18 ^a^	1.8 ± 0.56 ^b^
	2022	3.4 ± 0.51 ^c^	3.7 ± 0.06 ^b^	3.9 ± 1.29 ^b^	5.4 ± 0.76 ^c^
	2020	1.5 ± 0.30	1.7 ± 0.09 ^a^	1.3 ± 0.16 ^a^	1.0 ± 0.23 ^a^
Rosmarinic acid derivative	2021	2.8 ± 0.75	4.0 ± 0.82 ^b^	2.9 ± 0.58 ^b^	4.5 ± 1.23 ^b^
	2022	2.9 ± 0.91	3.7 ± 0.34 ^b^	3.3 ± 0.28 ^b^	3.7 ± 1.21 ^b^
	2020	0.04 ± 0.02 ^a^	0.1 ± 0.02	0.1 ± 0.01	0.1 ± 0.02 ^a^
Caffeic acid	2021	0.1 ± 0.02 ^b^	0.1 ± 0.02	0.1 ± 0.01	0.2 ± 0.01 ^b^
	2022	0.1 ± 0.02 ^b^	0.1 ± 0.03	0.1 ± 0.01	0.1 ± 0.01 ^a^
	2020	0.3 ± 0.08 ^a^	0.7 ± 0.02 ^a^	0.7 ± 0.14 ^a^	0.4 ± 0.10 ^a^
*p*-Coumaric acid glycoside	2021	1.4 ± 0.32 ^b^	1.5 ± 0.27 ^b^	0.8 ± 0.12 ^a^	1.4 ± 0.39 ^b^
	2022	1.5 ± 0.27 ^b^	1.7 ± 0.23 ^b^	1.6 ± 0.16 ^b^	1.6 ± 0.35 ^b^
	2020	0.3 ± 0.09	0.4 ± 0.08 ^a^	0.3 ± 0.04 ^a^	0.2 ± 0.07 ^a^
Ferulic acid hexoside	2021	0.3 ± 0.08	0.7 ± 0.13 ^b^	0.5 ± 0.09 ^b^	0.5 ± 0.06 ^b^
	2022	0.4 ± 0.07	0.9 ± 0.12 ^b^	0.6 ± 0.08 ^c^	0.6 ± 0.06 ^b^
	2020	0.3 ± 0.11 ^a^	0.3 ± 0.03 ^a^	0.3 ± 0.13 ^a^	0.3 ± 0.07 ^a^
Luteolin-7-*O*-glucoside	2021	0.5 ± 0.11 ^a^	0.4 ± 0.12 ^a^	0.4 ± 0.05 ^a^	0.5 ± 0.06 ^b^
	2022	0.8 ± 0.09 ^b^	0.8 ± 0.14 ^b^	0.7 ± 0.15 ^b^	0.8 ± 0.02 ^c^
	2020	0.9 ± 0.26 ^a^	1.1 ± 0.03 ^a^	1.0 ± 0.15 ^a^	0.8 ± 0.15 ^a^
Luteolin-7-*O*-glucuronide	2021	1.3 ± 0.25 ^ab^	1.4 ± 0.37 ^a^	1.1 ± 0.23 ^a^	1.4 ± 0.33 ^b^
	2022	1.9 ± 0.47 ^b^	2.5 ± 0.33 ^b^	2.2 ± 0.32 ^b^	2.3 ± 0.12 ^c^
	2020	0.3 ± 0.09 ^a^	0.4 ± 0.01	0.4 ± 0.08	0.3 ± 0.03
Apigenin-7-*O*-glucoside	2021	0.6 ± 0.10 ^b^	0.5 ± 0.14	0.4 ± 0.02	0.5 ± 0.07
	2022	0.6 ± 0.10 ^b^	0.7 ± 0.11	0.6 ± 0.04	0.5 ± 0.05
	2020	0.6 ± 0.17	0.7 ± 0.00 ^ab^	0.4 ± 0.21 ^a^	1.2 ± 0.43 ^b^
*o*-Coumaric acid	2021	0.7 ± 0.18	1.2 ± 0.37 ^b^	2.8 ± 0.48 ^b^	1.1 ± 0.53 ^a^
	2022	0.4 ± 0.19	0.3 ± 0.07 ^a^	0.3 ± 0.04 ^a^	0.4 ± 0.04 ^ba^
	2020	0.3 ± 0.15 ^a^	0.6 ± 0.01 ^a^	0.9 ± 0.31	0.6 ± 0.12 ^a^
Rosmarinic acid	2021	1.5 ± 0.09 ^c^	1.5 ± 0.08 ^b^	1.4 ± 0.01	1.4 ± 0.03 ^b^
	2022	0.9 ± 0.16 ^b^	1.5 ± 0.42 ^b^	1.3 ± 0.30	1.3 ± 0.20 ^b^
	2020	1.0 ± 0.26	1.1 ± 0.09 ^a^	0.7 ± 0.15 ^a^	0.7 ± 0.26 ^a^
Salvianolic acid A	2021	1.6 ± 0.62	2.1 ± 0.26 ^b^	1.2 ± 0.39 ^a^	1.9 ± 0.33 ^b^
	2022	1.9 ± 0.22	3.0 ± 0.92 ^c^	2.5 ± 0.48 ^b^	2.6 ± 0.16 ^c^
	2020	0.4 ± 0.06	0.4 ± 0.01 ^b^	0.4 ± 0.17 ^b^	0.5 ± 0.14 ^b^
Luteolin	2021	0.3 ± 0.10	0.4 ± 0.09 ^b^	1.6 ± 0.13 ^c^	0.3 ± 0.09 ^b^
	2022	0.2 ± 0.11	0.1 ± 0.02 ^a^	0.1 ± 0.01 ^a^	0.1 ± 0.01 ^a^
	2020	0.1 ± 0.01	0.1 ± 0.02	0.1 ± 0.01	0.1 ± 0.01
Salvianolic acid C	2021	0.1 ± 0.01	0.1 ± 0.03	0.1 ± 0.01	0.1 ± 0.01
	2022	0.1 ± 0.01	0.1 ± 0.02	0.1 ± 0.02	0.2 ± 0.02
	2020	0.1 ± 0.01 ^b^	0.1 ± 0.01 ^b^	0.1 ± 0.05 ^ab^	0.1 ± 0.03 ^b^
Apigenin	2021	0.1 ± 0.03 ^ab^	0.1 ± 0.02 ^b^	0.1 ± 0.04 ^b^	0.1 ± 0.04 ^b^
	2022	0.04 ± 0.0 ^a^	0.03 ± 0.00 ^a^	0.02 ± 0.00 ^a^	0.02 ± 0.00 ^a^

Results are expressed as mean ± SD. Different letters indicate a significant difference between years per component at *p* < 0.05.

## Data Availability

Data are available within this article.

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
