# Peer review of "Agronomic Evaluation and Chemical Characterization of Lavandula latifolia Medik. under the Semiarid Conditions of the Spanish Southeast"

_plants, 2023, doi:10.3390/plants12101986_

Round 1

Reviewer 1 Report

I have the following concerns about the scientific quality of presented paper:

1. The description of M&M is poor and incomplete. Many of used methods are significantly differs from the procedures described into cited references (4.4 Extraction of polyphenolic compounds; 4.5. HPLC-DAD analysis; 4.6 Antioxidant capacity (FRAP and DPPH) etc.);

2. The presented results for GC/MS analysis are incomplete (no data for RI was presented);

3. The presented results for HPLC-DAD quantifications are suspicious. The developed HPLC method, cited in M&M include quantification of only 5 compounds – here are quantified 14 compounds – I would like to see the chromatograms to confirm the right separation and the equations of standard curves, used for quantification;

4. The statement that “…the essential oil synthesized by this species is genetically predetermined…..” and that “…genetic factors, more than environ- 265 mental conditions, are directly related to the response of the secondary metabolism…” are speculative and must be experimentally proved;

5. The conclusions, especially “For the polyphenolic profile, and consequently for the antioxidant capacity of their extracts, an increase was observed between the first and the second year, but remained stable at the third, which suggests that after episodes of lack of water, the species could maintain this antioxidant system active over time.” are not supported by the results – almost all compounds and parameters continues to grow at the third year and no stationary  phase was reached and confirmed for at last three consecutive years to formulate such claim.

Reviewer 2 Report

Dear Authors,

The MS Agronomic evaluation and chemical characterization of La- vandula latifolia Medik. under the semiarid conditions of the Spanish Southeast.” is original research topic, well-written and can be accepted for publishing after minor revision.

Please, see some comments bellow

L76-77 These polyphenols are considered powerful metabolites with several applications owing to biological activity as mentioned before”. It would be good to add a sentence about correlation between presence of phenolics and antioxidant potential of extracts.

Ref:Pourreza N. Phenolic compounds as potential antioxidant. Jundishapur J Nat Pharm Prod. 2013 Nov;8(4):149-50. doi: 10.17795/jjnpp-15380.

L133-135 The chemotype was defined, as it was stated before, by three components: 1,8-cineole, linalool and camphor, based on the relative concentration at which these three major compounds were present in this volatile fraction.

It would be good to define or present another name of 1,8-cineole - eucalyptol 

Monoterpenoid 1,8-cineole, terpene linalool and camphor - terpenoid and a cyclic ketone. It would be good to say that chemotype was defined by presence terpene and terpenoids as further information about presence of α-pinene, β-pinene, and α-terpineol.

Reviewer 3 Report

The study provides interesting news about the possibility of using local spike lavender ecotypes. The manuscript needs same changes before the publication.

Line 27: standardise dot and comma.

Throughout the text: Names of family in Italic.

Introduction: this paragraph does not provide correct background. Many sentences are reported without references. “The main goal of the present study was to evaluate biomass production, essential oil (yield and quality), and richness of polyphenols from extracts of pre-selected spike lavender ecotypes, over three years of cultivation practices”, it might be useful to include information on the literature gaps.

Line 33-41: references.

Results

Table and figure: standardise captions. In figures, insert captions to better explain graphs content. For example FW.

Table 1: I suggest inserting on decimal number for EO content.

Table 2: I suggest inserting 3 lines (one for each year) with the total % of components identified.

Figure 2: improve the quality of the graphs.

Line 175-176: this phrase goes in discussion section.

Discussion: in this paragraph you can insert other information regarding results obtained by other authors in different areas.

Material and methods

Line 338: annual average rainfall or annual total rainfall?

Line 354-356: you say: “three consecutive harvests were made during the summers of 355 2020, 2021, and 2022, when the plant material showed a phenological stage of full blooms 356 and the beginning of fructification”. In the three years, did you observe this phenological stage at the same time? May be better to indicate harvest day or harvest week for each year.

Figure 3: improve quality of the graphs and standardise the year (june19-june 20 and june20/June 21). This graph shows that in the three years has always been found the same average temperature, but probably different trend was observed during each year. It may be useful to report the average monthly temperatures for each year.

You can insert a paragraph with cultivation practices adopted (fertilization, weed management)

Reviewer 4 Report

The study analyzed the agronomical production and phytochemical profile of two drought tolerant ecotypes of Lavandula latifolia Medik. under the semiarid conditions of the Spanish Southeast. After three years of cultivation practices, the biomass and essential oil productions were correlated with the age of plants and the essential oil chemotype was maintained over time.

The subject is important and the manuscript is well written. However, several suggestions should be addressed to improve the value of the manuscript; please see the followings:

English should be rechecked throughout

Acronyms/Abbreviations should be defined the first time they appear in the abstract, main text, and the first figure or table

Lines 57-65 - this idea can be added here: “the nature of soil may induce different molecule production secreted by microorganisms in the soil that stimulate and regulate the synthesis of EOs in medicinal plants (Chamkhi et al., 2021, doi: 10.1016/j.plaphy.2021.08.001). Moreover, EOs in lavander plants might vary between seasonal stages and plant parts (Angioni et al., 2006, doi: 10.1021/jf0603329).”

Lines 72-77 - this important finding can be added in this paragraph: “a recent in silico study revealed that some EOs from lavender could inhibit SARS-CoV-2 and might be possible candidates in the treatment of COVID-19 (Benali et al., 2023, doi: 10.3390/plants12061413).”

Line 92 - reference needed

Discussion: The text should not repeat data presented in Results but emphasize or summarize the most important observations and discuss/correlate them with previously published studies regarding plants from the same genus. The following articles are some examples:

Carrasco et al., 2016, doi: 10.1080/14786419.2015.1043632

Carrasco et al., 2015, doi: 10.1016/j.indcrop.2015.03.088

Insawang et al., 2019, doi: 10.1002/cbdv.201900371

Biltekin et al., 2022, doi: 10.1021/acsomega.2c04518

Conclusion - Authors should mention the limitations of their study and the scope for future research

Reviewer 5 Report

It is opinion of the reviewer that this paper before acceptance needs several corrections. My individual comments are listed below.

L. 1 - The title should be written with capital letters.

L. 7-11 – The authors’ initials and e-mail addresses should be completed.

The Abstract should provide more information about antioxidant activity.

L. 32-33 – It should be “…serve producing …commercial crops [1].”

L. 73 – Hydroxybenzoic and hydroxycinnamic acids are the simple phenolics and cannot be called as polyphenols. I suggest to use in entire paper a term of “phenolic compounds” or “phenolics” instead of “polyphenolics/polyphenols”.

Figure 1 – In Y axis description, it should be “kg” instead of “Mg”.

Table 3, page 8 – It should be “o-Coumaric acid”.

Table 3 – It should be “Phenolic …” instead of “Polyphenolic ….”.

Table 2 & 3 –Information about significance of differences between mean values should be presented in footnote not in the title.

L. 172 – “-O-“ should be written with italic.

L. 177 – It should be “o-coumaric”.

L. 205 – It should be “antioxidant potential”.

L. 207 – It should be “assays” instead of “techniques”.

L. 207, Figure 2 Y axis description – It should be “TE” not “TEAC”.

Figure 2 – It should be “….DPPH assay  …. FRAP assay …”.

L. 367 – It should be “Essential oils were extracted according to protocol …”.

L. 391 – It should be “… were expressed according to …”.

L. 413 – It should be “UV-Vis….”.

L. 421 – It should be “as mg per g of …”.

L. 424 – It should be “(FRAP and DPPH assays)”.

L. 427 – It should be “-s-“ with italic.

L. 432 – It should be “DPPH radical” or “DPPH•”.

L. 431 – It should be “µmol Fe2+/g…”.

References – The Latin names must be written with italic. The second part of the Latin name must be written with lower case letter – for example  “Lavendula latifolia”, “Salvia officinalis”, “Rosmarinus officialis”.

Round 2

Reviewer 4 Report

The authors answered the questions and addressed the suggestions resulting in a much improved manuscript.

I have no further comments.